# Auto-Encoding Sequential Monte Carlo

**Tuan Anh Le**[†]**, Maximilian Igl**[†]**, Tom Rainforth**[‡]**, Tom Jin**[†,§]**, Frank Wood**[†]

[†]Department of Engineering Science, University of Oxford
[‡]Department of Statistics, University of Oxford
[§]Department of Statistics, University of Warwick
`{tuananh,igl,jin,fwood}@robots.ox.ac.uk,    rainforth@stats.ox.ac.uk`

## Abstract

We build on auto-encoding sequential Monte Carlo (AESMC):[1] a method for model and proposal learning based on maximizing the lower bound to the log marginal likelihood in a broad family of structured probabilistic models. Our approach relies on the efficiency of sequential Monte Carlo (SMC) for performing inference in structured probabilistic models and the flexibility of deep neural networks to model complex conditional probability distributions. We develop additional theoretical insights and experiment with a new training procedure which can improve both model and proposal learning. We demonstrate that our approach provides a fast, easy-to-implement and scalable means for simultaneous model learning and proposal adaptation in deep generative models.

## 1 Introduction

We build upon AESMC (Le et al., 2017), a method for model learning that itself builds on variational auto-encoders (VAEs) (Kingma & Welling, 2014; Rezende et al., 2014) and importance weighted auto-encoders (IWAEs) (Burda et al., 2016). AESMC is similarly based on maximizing a lower bound to the log marginal likelihood, but uses SMC (Doucet & Johansen, 2009) as the underlying marginal likelihood estimator instead of importance sampling (IS). For a very wide array of models, particularly those with sequential structure, SMC forms a substantially more powerful inference method than IS, typically returning lower variance estimates for the marginal likelihood. Consequently, by using SMC for its marginal likelihood estimation, AESMC often leads to improvements in model learning compared with VAEs and IWAEs. We provide experiments on structured time-series data that show that AESMC based learning was able to learn useful representations of the latent space for both reconstruction and prediction more effectively than the IWAE counterpart.

AESMC was introduced in an earlier preprint (Le et al., 2017) concurrently with the closely related methods of Maddison et al. (2017); Naesseth et al. (2017). In this work we take these ideas further by providing new theoretical insights for the resulting evidence lower bounds (ELBOs), extending these to explore the relative efficiency of different approaches to proposal learning, and using our results to develop a new and improved training procedure. In particular, we introduce a method for expressing the gap between an ELBO and the log marginal likelihood as a Kullback-Leibler (KL) divergence between two distributions on an extended sampling space. Doing so allows us to investigate the behavior of this family of algorithms when the objective is maximized perfectly, which occurs only if the KL divergence becomes zero. In the IWAE case, this implies that the proposal distributions are equal to the posterior distributions under the learned model. In the AESMC case, it has implications for both the proposal distributions and the intermediate set of targets that are learned. We demonstrate that, somewhat counter-intuitively, using lower variance estimates for the marginal likelihood can actually be harmful to proposal learning. Using these insights, we experiment with an adaptation to the AESMC algorithm, which we call *alternating* ELBOs, that uses different lower bounds for updating the model parameters and proposal parameters. We observe that this adaptation can, in some cases, improve model learning and proposal adaptation.

---

[1]This work builds upon an earlier preprint (Le et al., 2017) along with the independent, simultaneously developed, closely related, work of Maddison et al. (2017) and Naesseth et al. (2017).

## 2 BACKGROUND

### 2.1 STATE-SPACE MODELS

State-space models (SSMs) are probabilistic models over a set of latent variables $x_{1:T}$ and observed variables $y_{1:T}$. Given parameters $\theta$, a SSM is characterized by an initial density $\mu_\theta(x_1)$, a series of transition densities $f_{t,\theta}(x_t|x_{1:t-1})$, and a series of emission densities $g_{t,\theta}(y_t|x_{1:t})$ with the joint density being $p_\theta(x_{1:T}, y_{1:T}) = \mu_\theta(x_1) \prod_{t=2}^T f_{t,\theta}(x_t|x_{1:t-1}) \prod_{t=1}^T g_{t,\theta}(y_t|x_{1:t})$.

We are usually interested in approximating the posterior $p_\theta(x_{1:T}|y_{1:T})$ or the expectation of some test function $\varphi$ under this posterior $I(\varphi) := \int \varphi(x_{1:T})p_\theta(x_{1:T}|y_{1:T}) \, dx_{1:T}$. We refer to these two tasks as inference. Inference in models which are non-linear, non-discrete, and non-Gaussian is difficult and one must resort to approximate methods, for which SMC has been shown to be one of the most powerful approaches (Doucet & Johansen, 2009).

We will consider model learning as a problem of maximizing the marginal likelihood $p_\theta(y_{1:T}) = \int p_\theta(x_{1:T}, y_{1:T}) \, dx_{1:T}$ in the family of models parameterized by $\theta$.

### 2.2 SEQUENTIAL MONTE CARLO

SMC performs approximate inference on a sequence of target distributions $(\pi_t(x_{1:t}))_{t=1}^T$. In the context of SSMs, the target distributions are often taken to be $(p_\theta(x_{1:t}|y_{1:t}))_{t=1}^T$. Given a parameter $\phi$ and proposal distributions $q_{1,\phi}(x_1|y_1)$ and $(q_{t,\phi}(x_t|y_{1:t}, x_{1:t-1}))_{t=2}^T$ from which we can sample and whose densities we can evaluate, SMC is described in Algorithm 1.

Using the set of weighted particles $(\tilde{x}_{1:T}^k, w_T^k)_{k=1}^K$ at the last time step, we can approximate the posterior as $\sum_{k=1}^K \bar{w}_T^k \delta_{\tilde{x}_{1:T}^k}(x_{1:T})$ and the integral $I_\varphi$ as $\sum_{k=1}^K \bar{w}_T^k \varphi(\tilde{x}_{1:T}^k)$, where $\bar{w}_T^k := w_T^k / \sum_j w_T^j$ is the normalized weight and $\delta_z$ is a Dirac measure centered on $z$. Furthermore, one can obtain an unbiased estimator of the marginal likelihood $p_\theta(y_{1:T})$ using the intermediate particle weights:

$$\hat{Z}_{\text{SMC}} := \prod_{t=1}^T \left[ \frac{1}{K} \sum_{k=1}^K w_t^k \right]. \tag{1}$$

---

**Algorithm 1:** Sequential Monte Carlo

**Data:** observed values $y_{1:T}$, model parameters $\theta$, proposal parameters $\phi$

**begin**

    Sample initial particle values $x_1^k \sim q_{1,\phi}(\cdot|y_1)$.

    Compute and normalize weights:

$$w_1^k = \frac{\mu_\theta(x_1^k) g_{1,\theta}(y_1|x_1^k)}{q_{1,\phi}(x_1^k|y_1)}, \qquad \bar{w}_1^k = \frac{w_1^k}{\sum_{\ell=1}^K w_1^\ell}.$$

    Initialize particle set: $\tilde{x}_1^k \leftarrow x_1^k$

    **for** $t = 2, 3, \ldots, T$ **do**

        Sample ancestor index $a_{t-1}^k \sim \text{Discrete}(\cdot|\bar{w}_{t-1}^1, \ldots, \bar{w}_{t-1}^K)$.

        Sample particle value $x_t^k \sim q_{t,\phi}(\cdot|y_{1:t}, \tilde{x}_{1:t-1}^{a_{t-1}^k})$.

        Update particle set $\tilde{x}_{1:t}^k \leftarrow (\tilde{x}_{1:t-1}^{a_{t-1}^k}, x_t^k)$.

        Compute and normalize weights:

$$w_t^k = \frac{f_{t,\theta}(x_t^k|\tilde{x}_{1:t-1}^{a_{t-1}^k}) g_{t,\theta}(y_t|\tilde{x}_{1:t}^k)}{q_{t,\phi}(x_t^k|y_{1:t}, \tilde{x}_{1:t-1}^{a_{t-1}^k})}, \qquad \bar{w}_t^k = \frac{w_t^k}{\sum_{\ell=1}^K w_t^\ell}.$$

    Compute marginal likelihood: $\hat{Z}_{\text{SMC}} = \prod_{t=1}^T \frac{1}{K} \sum_{k=1}^K w_t^k$.

**return** *particles* $(\tilde{x}_{1:T}^k)_{k=1}^K$, *weights* $(w_T^k)_{k=1}^K$, *marginal likelihood estimate* $\hat{Z}_{SMC}$

---

The sequential nature of SMC and the resampling step are crucial in making SMC scalable to large $T$. The former makes it easier to design efficient proposal distributions as each step need only target the next set of variables $x_t$. The resampling step allows the algorithm to focus on promising particles in light of new observations, avoiding the exponential divergence between the weights of different samples that occurs for importance sampling as $T$ increases. This can be demonstrated both empirically and theoretically (Del Moral, 2004, Chapter 9). We refer the reader to (Doucet & Johansen, 2009) for an in-depth treatment of SMC.

### 2.3 IMPORTANCE WEIGHTED AUTO-ENCODERS

Given a dataset of observations $(y^{(n)})_{n=1}^N$, a generative network $p_\theta(x, y)$ and an inference network $q_\phi(x|y)$, IWAEs (Burda et al., 2016) maximize $\frac{1}{N} \sum_{n=1}^N \text{ELBO}_{\text{IS}}(\theta, \phi, y^{(n)})$ where, for a given observation $y$, the ELBO$_{\text{IS}}$ (with $K$ particles) is a lower bound on $\log p_\theta(y)$ by Jensen's inequality:

$$\text{ELBO}_{\text{IS}}(\theta, \phi, y) = \int Q_{\text{IS}}(x^{1:K}) \log \hat{Z}_{\text{IS}}(x^{1:K}) \, \mathrm{d}x^{1:K} \leq \log p_\theta(y), \text{ where} \tag{2}$$

$$Q_{\text{IS}}(x^{1:K}) = \prod_{k=1}^K q_\phi(x^k|y), \quad \hat{Z}_{\text{IS}}(x^{1:K}) = \frac{1}{K} \sum_{k=1}^K \frac{p_\theta(x^k, y)}{q_\phi(x^k|y)}. \tag{3}$$

Note that for $K = 1$ particle, this objective reduces to a VAE (Kingma & Welling, 2014; Rezende et al., 2014) objective we will refer to as

$$\text{ELBO}_{\text{VAE}}(\theta, \phi, y) = \int q_\phi(x|y)(\log p_\theta(x, y) - \log q_\phi(x|y)) \, \mathrm{d}x. \tag{4}$$

The IWAE optimization is performed using stochastic gradient ascent (SGA) where a sample from $\left(\prod_{k=1}^K q_\phi(x^k|y^{(n)})\right)$ is obtained using the reparameterization trick (Kingma & Welling, 2014) and the gradient $\frac{1}{N} \sum_{n=1}^N \nabla_{\theta, \phi} \log \left(\sum_{k=1}^K \frac{p_\theta(x^k, y^{(n)})}{q_\phi(x^k|y^{(n)})}\right)$ is used to perform an optimization step.

## 3 AUTO-ENCODING SEQUENTIAL MONTE CARLO

AESMC implements model learning, proposal adaptation, and inference amortization in a similar manner to the VAE and the IWAE: it uses SGA on an empirical average of the ELBO over observations. However, it varies in the form of this ELBO. In this section, we will introduce the AESMC ELBO, explain how gradients of it can be estimated, and discuss the implications of these changes.

### 3.1 OBJECTIVE FUNCTION

Consider a family of SSMs $\{p_\theta(x_{1:T}, y_{1:T}) : \theta \in \Theta\}$ and a family of proposal distributions $\{q_\phi(x_{1:T}|y_{1:T}) = q_{1,\phi}(x_1|y_1) \prod_{t=2}^T q_{t,\phi}(x_t|x_{1:t-1}, y_{1:t}) : \phi \in \Phi\}$. AESMC uses an ELBO objective based on the SMC marginal likelihood estimator (1). In particular, for a given $y_{1:T}$, the objective is defined as

$$\text{ELBO}_{\text{SMC}}(\theta, \phi, y_{1:T}) := \int Q_{\text{SMC}}(x_{1:T}^{1:K}, a_{1:T-1}^{1:K}) \log \hat{Z}_{\text{SMC}}(x_{1:T}^{1:K}, a_{1:T-1}^{1:K}) \, \mathrm{d}x_{1:T}^{1:K} \, \mathrm{d}a_{1:T-1}^{1:K}, \tag{5}$$

where $\hat{Z}_{\text{SMC}}(x_{1:T}^{1:K}, a_{1:T-1}^{1:K})$ is defined in (1) and $Q_{\text{SMC}}$ is the sampling distribution of SMC,

$$Q_{\text{SMC}}(x_{1:T}^{1:K}, a_{1:T-1}^{1:K}) = \left(\prod_{k=1}^K q_{1,\phi}(x_1^k)\right) \left(\prod_{t=2}^T \prod_{k=1}^K q_{t,\phi}(x_t^k|\tilde{x}_{1:t-1}^{a_{t-1}^k}) \cdot \text{Discrete}(a_{t-1}^k|w_{t-1}^{1:K})\right). \tag{6}$$

ELBO$_{\text{SMC}}$ forms a lower bound to the log marginal likelihood $\log p_\theta(y_{1:T})$ due to Jensen's inequality and the unbiasedness of the marginal likelihood estimator. Hence, given a dataset $(y_{1:T}^{(n)})_{n=1}^N$, we can perform model learning based on maximizing the lower bound of $\frac{1}{N} \sum_{n=1}^N \log p_\theta(y_{1:T}^{(n)})$ as a surrogate target, namely by maximizing

$$\mathcal{J}(\theta, \phi) := \frac{1}{N} \sum_{n=1}^N \text{ELBO}_{\text{SMC}}(\theta, \phi, y_{1:T}^{(n)}). \tag{7}$$

For notational convenience, we will talk about optimizing ELBOs in the rest of this section. However, we note that the main intended use of AESMC is to amortize over datasets, for which the ELBO is replaced by the dataset average $\mathcal{J}(\theta, \phi)$ in the optimization target. Nonetheless, rather than using the full dataset for each gradient update, will we instead use minibatches, noting that this forms unbiased estimator.

## 3.2 GRADIENT ESTIMATION

We describe a gradient estimator used for optimizing $\text{ELBO}_{\text{SMC}}(\theta, \phi, y_{1:T})$ using SGA. The SMC sampler in Algorithm 1 proceeds by sampling $x_1^{1:K}, a_1^{1:K}, x_2^{1:K}, \dots$ sequentially from their respective distributions $\prod_{k=1}^{K} q_1(x_1^k)$, $\prod_{k=1}^{K} \text{Discrete}(a_1^k | w_1^{1:K})$, $\prod_{k=1}^{K} q_2(x_2^k | x_1^{a_1^k}), \dots$ until the whole particle-weight trajectory $(x_{1:K}^{1:T}, a_{1:T-1}^{1:K})$ is sampled. From this trajectory, using equation (1), we can obtain an estimator for the marginal likelihood.

Assuming that the sampling of latent variables $x_{1:T}^{1:K}$ is reparameterizable, we can make their sampling independent of $(\theta, \phi)$. In particular, assume that there exists a set of auxiliary random variables $\epsilon_{1:T}^{1:K}$ where $\epsilon_t^k \sim s_t$ and a set of reparameterization functions $r_t$. We can simulate the SMC sampler by first sampling $\epsilon_1^{1:K} \sim \prod_{k=1}^{K} s_1$ and setting $x_1^k = r_1(\epsilon_1^k)$ and $\tilde{x}_1^k = x_1^k$, then for $t = 2, \dots, T$ cycling through sampling $a_{t-1}^{1:K} \sim \prod_{k=1}^{K} \text{Discrete}(a_{t-1}^k | w_{t-1}^{1:K})$ and $\epsilon_t^{1:K} \sim \prod_{k=1}^{K} s_t$, and setting $x_t^k = r_t(\epsilon_t^k, \tilde{x}_{1:t-1}^{a_{t-1}^k})$ and $\tilde{x}_{1:t}^k = (\tilde{x}_{1:t-1}^{a_{t-1}^k}, x_t^k)$. We use the resulting reparameterized sample of $(x_{1:K}^{1:T}, a_{1:T-1}^{1:K})$ to evaluate the gradient estimator $\nabla_{\theta, \phi} \log \hat{Z}_{\text{SMC}}(x_{1:T}^{1:K}, a_{1:T-1}^{1:K})$.

To account for the discrete choices of ancestor indices $a_t^k$ one could additionally use the REINFORCE (Williams, 1992) trick, however in practice, we found that the additional term in the estimator has problematically high variance. We explore various other possible gradient estimators and empirical assessments of their variances in Appendix A. This exploration confirms that including the additional REINFORCE terms leads to problematically high variance, justifying our decision to omit them, despite introducing a small bias into the gradient estimates.

## 3.3 BIAS & IMPLICATIONS ON THE PROPOSALS

In this section, we express the gap between ELBOs and the log marginal likelihood as a KL divergence and study implications on the proposal distributions. We present a set of claims and propositions whose full proofs are in Appendix B. These give insight into the behavior of AESMC and show the advantages, and disadvantages, of using our different ELBO. This insight motivates Section 4 which proposes an algorithm for improving proposal learning.

**Definition 1.** *Given an* unnormalized target density $\tilde{P} : \mathcal{X} \to [0, \infty)$ *with* normalizing constant $Z_P > 0$, $P := \tilde{P}/Z_P$, *and a* proposal density $Q : \mathcal{X} \to [0, \infty)$, *then*

$$\text{ELBO} := \int Q(x) \log \frac{\tilde{P}(x)}{Q(x)} \, \mathrm{d}x, \tag{8}$$

*is a lower bound on* $\log Z_P$ *and satisfies*

$$\text{ELBO} = \log Z_P - \text{KL}\left(Q || P\right). \tag{9}$$

This is a standard identity used in variational inference and VAEs. In the case of VAEs, applying Definition 1 with $P$ being $p_\theta(x|y)$, $\tilde{P}$ being $p_\theta(x, y)$, $Z_P$ being $p_\theta(y)$, and $Q$ being $q_\phi(x|y)$, we can directly rewrite (4) as $\text{ELBO}_{\text{VAE}}(\theta, \phi, y) = \log p_\theta(y) - \text{KL}\left(q_\phi(x|y) || p_\theta(x|y)\right)$.

The key observation for expressing such a bound for general ELBOs such as $\text{ELBO}_{\text{IS}}$ and $\text{ELBO}_{\text{SMC}}$ is that the target density $P$ and the proposal density $Q$ need not directly correspond to $p_\theta(x|y)$ and $q_\phi(x|y)$. This allows us to view the underlying sampling distributions of the marginal likelihood Monte Carlo estimators such as $Q_{\text{IS}}$ in (3) and $Q_{\text{SMC}}$ in (6) as proposal distributions on an extended space $\mathcal{X}$. The following claim uses this observation to express the bound between a general ELBO and the log marginal likelihood as KL divergence from the extended space sampling distribution to a corresponding target distribution.

**Claim 1.** *Given a non-negative unbiased estimator $\hat{Z}_P(x) \geq 0$ of the normalizing constant $Z_P$ where $x$ is distributed according to the proposal distribution $Q(x)$, the following holds:*

$$\text{ELBO} = \int Q(x) \log \hat{Z}_P(x) \, \mathrm{d}x = \log Z_P - \text{KL}\left(Q||P\right), \tag{10}$$

$$\text{where} \quad P(x) = \frac{Q(x)\hat{Z}_P(x)}{Z_P} \tag{11}$$

*is the implied normalized target density.*

In the case of IWAEs, we can apply Claim 1 with $Q$ and $\hat{Z}_P$ being $Q_{\text{IS}}$ and $\hat{Z}_{\text{IS}}$ respectively as defined in (3) and $Z_P$ being $p_\theta(y)$. This yields

$$\text{ELBO}_{\text{IS}}(\theta, \phi, y) = \log p_\theta(y) - \text{KL}\left(Q_{\text{IS}}||P_{\text{IS}}\right), \text{ where} \tag{12}$$

$$P_{\text{IS}}(x^{1:K}) = \frac{1}{K} \sum_{k=1}^{K} \left(q_\phi(x^1|y) \cdots q_\phi(x^{k-1}|y) p_\theta(x^k|y) q_\phi(x^{k+1}|y) \cdots q_\phi(x^K|y)\right). \tag{13}$$

Similarly, in the case of AESMC, we obtain

$$\text{ELBO}_{\text{SMC}}(\theta, \phi, y_{1:T}) = \log p_\theta(y_{1:T}) - \text{KL}\left(Q_{\text{SMC}}||P_{\text{SMC}}\right), \text{ where} \tag{14}$$

$$P_{\text{SMC}}(x_{1:T}^{1:K}, a_{1:T-1}^{1:K}) = Q_{\text{SMC}}(x_{1:T}^{1:K}, a_{1:T-1}^{1:K}) \hat{Z}_{\text{SMC}}(x_{1:T}^{1:K}, a_{1:T-1}^{1:K})/p_\theta(y_{1:T}). \tag{15}$$

Having expressions for the target distribution $P$ and the sampling distribution $Q$ for a given ELBO allows us to investigate what happens when we maximize that ELBO, remembering that the KL term is strictly non-negative and zero if and only if $P = Q$. For the VAE and IWAE cases then, provided the proposal is sufficiently flexible, one can always perfectly maximize the ELBO by setting $p_\theta(x|y) = q_\phi(x|y)$ for all $x$. The reverse implication also holds: if $\text{ELBO}_{\text{VAE}} = \log Z_P$ then it must be the case that $p_\theta(x|y) = q_\phi(x|y)$. However, for AESMC, achieving $\text{ELBO} = \log Z_P$ is only possible when one also has sufficient flexibility to learn a particular series of intermediate target distributions, namely the marginals of the final target distribution. In other words, it is necessary to learn a particular factorization of the generative model, not just the correct individual proposals, to achieve $P = Q$ and thus $\text{ELBO}_{\text{SMC}} = Z_P$. These observations are formalized in Propositions 1 and 2 below.

**Proposition 1.** $Q_{\text{IS}}(x^{1:K}) = P_{\text{IS}}(x^{1:K})$ *for all $x^{1:K}$ if and only if $q(x|y) = p(x|y)$ for all $x$.*

**Proposition 2.** *If $K > 1$, then $P_{\text{SMC}}(x_{1:T}^{1:K}, a_{1:T-1}^{1:K}) = Q_{\text{SMC}}(x_{1:T}^{1:K}, a_{1:T-1}^{1:K})$ for all $(x_{1:T}^{1:K}, a_{1:T-1}^{1:K})$ if and only if*

1. *$\pi_t(x_{1:t}) = \int p(x_{1:T}|y_{1:T}) \, \mathrm{d}x_{t+1:T} = p(x_{1:t}|y_{1:T})$ for all $x_{1:t}$ and $t = 1, \ldots, T$, and*

2. *$q_1(x_1|y_1) = p(x_1|y_{1:T})$ for all $x_1$ and $q_t(x_t|x_{1:t-1}, y_{1:t}) = p(x_{1:t}|y_{1:T})/p(x_{1:t-1}|y_{1:T})$ for $t = 2, \ldots, T$ for all $x_{1:t}$,*

*where $\pi_t(x_{1:t})$ are the intermediate targets used by SMC.*

Proposition 2 has the consequence that if the family of generative models is such that the first condition does not hold, we will not be able to make the bound tight. This means that, except for a very small class of models, then, for most convenient parameterizations, it will be impossible to learn a perfect proposal that gives a tight bound, i.e. there will be no $\theta$ and $\phi$ such that the above conditions can be satisfied. However, it also means that $\text{ELBO}_{\text{SMC}}$ encodes important additional information about the implications the factorization of the generative model has on the inference—the model depends only on the final target $\pi_T(x_{1:T}) = p_\theta(x_{1:T}|y_{1:T})$, but some choices of the intermediate targets $\pi_t(x_{1:t})$ will lead to much more efficient inference than others. Perhaps more importantly, SMC is usually a far more powerful inference algorithm than importance sampling and so the AESMC setup allows for more ambitious model learning problems to be effectively tackled than the VAE or IWAE. After all, even though it is well known in the SMC literature that, unlike for IS, most problems have no perfect set of SMC proposals which will generate exact samples from the posterior (Doucet & Johansen, 2009), SMC still gives superior performance on most problems with more than a few dimensions. These intuitions are backed up by our experiments that show that using $\text{ELBO}_{\text{SMC}}$ regularly learns better models than using $\text{ELBO}_{\text{IS}}$.

## 4  IMPROVING PROPOSAL LEARNING

In practice, one is rarely able to perfectly drive the divergence to zero and achieve a perfect proposal. In addition to the implications of the previous section, this occurs because $q_\phi(x_{1:T}|y_{1:T})$ may not be sufficiently expressive to represent $p_\theta(x_{1:T}|y_{1:T})$ exactly and because of the inevitable sub-optimality of the optimization process, remembering that we are aiming to learn an amortized inference artifact, rather than a single posterior representation. Consequently, to accurately assess the merits of different ELBOS for proposal learning, it is necessary to consider their finite-time performance. We therefore now consider the effect the number of particles $K$ has on the gradient estimators for $\text{ELBO}_{\text{IS}}$ and $\text{ELBO}_{\text{SMC}}$.

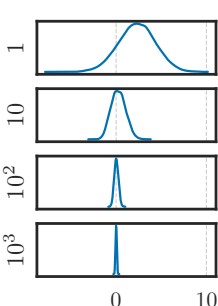

Figure 1: Density estimate of $\nabla_\phi \text{ELBO}$ for different $K$

Counter-intuitively, it transpires that the tighter bounds implied by using a larger $K$ is often harmful to proposal learning for both IWAE and AESMC. At a high-level, this is because an accurate estimate for $\hat{Z}_P$ can be achieved for a wide range of proposal parameters $\phi$ and so the magnitude of $\nabla_\phi \text{ELBO}$ reduces as $K$ increases. Typically, this shrinkage happens faster than increasing $K$ reduces the standard deviation of the estimate and so the standard deviation of the gradient estimate relative to the problem scaling (i.e. as a ratio of true gradient $\nabla_\phi \text{ELBO}$) actually increases. This effect is demonstrated in Figure 1 which shows a kernel density estimator for the distribution of the gradient estimate for different $K$ and the model given in Section 5.2. Here we see that as we increase $K$, both the expected gradient estimate (which is equal to the true gradient by unbiasedness) and standard deviation of the estimate decrease. However, the former decreases faster and so the relative standard deviation increases. This is perhaps easiest to appreciate by noting that for $K > 10$, there is a roughly equal probability of the estimate being positive or negative, such that we are equally likely to increase or decrease the parameter value at the next SGA iteration, inevitably leading to poor performance. On the other hand, when $K = 1$, it is far more likely that the gradient estimate is positive than negative, and so there is clear drift to the gradient steps. We add to the empirical evidence for this behavior in Section 5. Note the critical difference for model learning is that $\nabla_\theta \text{ELBO}$ does not, in general, decrease in magnitude as $K$ increases. Note also that using a larger $K$ should always give better performance at test time; it may though be better to learn $\phi$ using a smaller $K$.

In simultaneously developed work (Rainforth et al., 2017), we formalized this intuition in the IWAE setting by showing that the estimator of $\nabla_\phi \text{ELBO}_{\text{IS}}(\theta, \phi, x)$ with $K$ particles, denoted by $I_K$, has the following signal-to-noise ratio (SNR):

$$\text{SNR} := \frac{\mathbb{E}[I_K]}{\sqrt{\text{Var}[I_K]}} = O\left(\sqrt{\frac{1}{K}}\right). \tag{16}$$

We thus see that increasing $K$ reduces the SNR and so the gradient updates for the proposal will degrade towards pure noise if $K$ is set too high.

### 4.1  ALTERNATING ELBOS

To address these issues, we suggest and investigate the alternating ELBOS (ALT) algorithm which updates $(\theta, \phi)$ in a coordinate descent fashion using different ELBOS, and thus gradient estimates, for each. We pick a $\theta$-optimizing pair and a $\phi$-optimizing pair $(A_\theta, K_\theta), (A_\phi, K_\phi) \in \{\text{IS}, \text{SMC}\} \times \{1, 2, \dots\}$, corresponding to an inference type and number of particles. In an optimization step, we obtain an estimator for $\nabla_\theta \text{ELBO}_{A_\theta}$ with $K_\theta$ particles and an estimator for $\nabla_\phi \text{ELBO}_{A_\phi}$ with $K_\phi$ particles which we call $g_\theta$ and $g_\phi$ respectively. We use $g_\theta$ to update the current $\theta$ and $g_\phi$ to update the current $\phi$. The results from the previous sections suggest that using $A_\theta = \text{SMC}$ and $A_\phi = \text{IS}$ with a large $K_\theta$ and a small $K_\phi$ may perform better model and proposal learning than just fixing $(A_\theta, K_\theta) = (A_\phi, K_\phi)$ to (SMC, large) since using $A_\phi = \text{IS}$ with small $K_\phi$ helps learning $\phi$ (at least in terms of the SNR) and using $A_\theta = \text{SMC}$ with large $K_\theta$ helps learning $\theta$. We experimentally observe that this procedure can in some cases improve both model and proposal learning.

## 5 EXPERIMENTS

We now present a series of experiments designed to answer the following questions: 1) Does tightening the bound by using either more particles or a better inference procedure lead to an adverse effect on proposal learning? 2) Can AESMC, despite this effect, outperform IWAE? 3) Can we further improve the learned model and proposal by using ALT?

First we investigate a linear Gaussian state space model (LGSSM) for model learning and a latent variable model for proposal adaptation. This allows us to compare the learned parameters to the optimal ones. Doing so, we confirm our conclusions for this simple problem.

We then extend those results to more complex, high dimensional observation spaces that require models and proposals parameterized by neural networks. We do so by investigating the *Moving Agents* dataset, a set of partially occluded video sequences.

### 5.1 LINEAR GAUSSIAN STATE SPACE MODEL

Given the following LGSSM

$$p(x_1) = \text{Normal}\left(x_1; 0, 1^2\right), \tag{17}$$

$$p(x_t|x_{t-1}) = \text{Normal}\left(x_t; \theta_1 x_{t-1}, 1^2\right), \qquad t = 2, \ldots T, \tag{18}$$

$$p(y_t|x_t) = \text{Normal}\left(y_t; \theta_2 x_t, \sqrt{0.1}^2\right), \qquad t = 1, \ldots, T, \tag{19}$$

we find that optimizing $\text{ELBO}_{\text{SMC}}(\theta, \phi, y_{1:T})$ w.r.t. $\theta$ leads to better generative models than optimizing $\text{ELBO}_{\text{IS}}(\theta, \phi, y_{1:T})$. The same is true for using more particles.

We generate a sequence $y_{1:T}$ for $T = 200$ by sampling from the model with $\theta = (\theta_1, \theta_2) = (0.9, 1.0)$. We then optimize the different ELBOs w.r.t. $\theta$ using the bootstrap proposal $q_1(x_1|y_1) = \mu_\theta(x_1)$ and $q_t(x_t|x_{1:t-1}, y_{1:t}) = f_{t,\theta}(x_t|x_{1:t-1})$. Because we use the bootstrap proposal, gradients w.r.t. to $\theta$ are not backpropagated through $q$.

We use a fixed learning rate of 0.01 and optimize for 500 steps using SGA. Figure 2 shows that the convergence of both $\log p_\theta(y_{1:T})$ to $\max_\theta \log p_\theta(y_{1:T})$ and $\theta$ to $\text{argmax}_\theta \log p_\theta(y_{1:T})$ is faster when $\text{ELBO}_{\text{SMC}}$ and more particles are used.

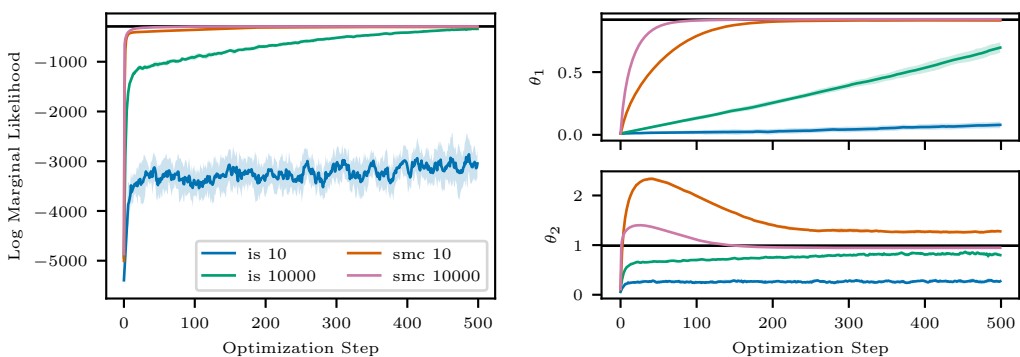

Figure 2: (Left) Log marginal likelihood analytically evaluated at every $\theta$ during optimization; the black line indicates $\max_\theta \log p_\theta(y_{1:T})$ obtained by the expectation maximization (EM) algorithm. (Right) learning of model parameters; the black line indicates $\text{argmax}_\theta \log p_\theta(y_{1:T})$ obtained by the EM algorithm.

### 5.2 PROPOSAL LEARNING

We now investigate how learning $\phi$, i.e. the proposal, is affected by the the choice of ELBO and the number of particles.

Consider a simple, fixed generative model $p(\mu)p(x|\mu) = \mathrm{Normal}(\mu; 0, 1^2)\mathrm{Normal}(x; \mu, 1^2)$ where $\mu$ and $x$ are the latent and observed variables respectively and a family of proposal distributions $q_\phi(\mu) = \mathrm{Normal}(\mu; \mu_q, \sigma_q^2)$ parameterized by $\phi = (\mu_q, \log \sigma_q^2)$. For a fixed observation $x = 2.3$, we initialize $\phi = (0.01, 0.01)$ and optimize $\mathrm{ELBO_{IS}}$ with respect to $\phi$. We investigate the quality of the learned parameter $\phi$ as we increase the number of particles $K$ during training. Figure 3 (left) clearly demonstrates that the quality of $\phi$ compared to the analytic posterior decreases as we increase $K$.

Similar behavior is observed in Figure 3 (middle, right) where we optimize $\mathrm{ELBO_{SMC}}$ with respect to both $\theta$ and $\phi$ for the LGSSM described in Section 5.1. We see that using more particles helps model learning but makes proposal learning worse. Using our ALT algorithm alleviates this problem and at the same time makes model learning faster as it profits from a more accurate proposal distribution. We provide more extensive experiments exploring proposal learning with different ELBOs and number of particles in Appendix C.3.

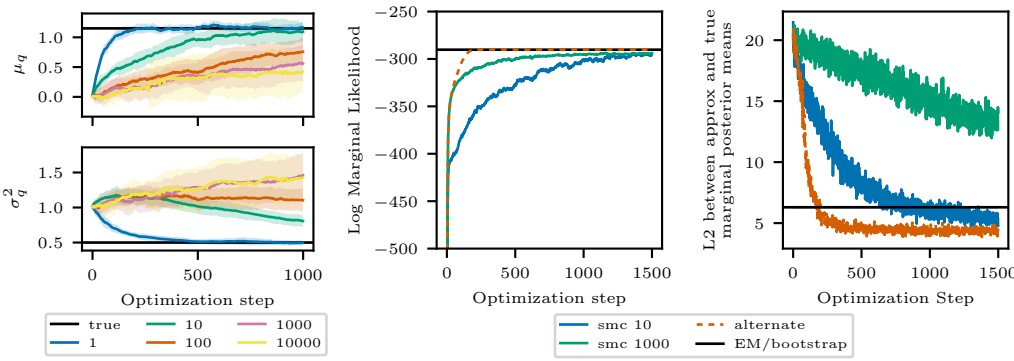

Figure 3: *(Left)* Optimizing $\mathrm{ELBO_{IS}}$ for the Gaussian unknown mean model with respect to $\phi$ results in worse $\phi$ as we increase number of particles $K$. *(Middle, right)* Optimizing $\mathrm{ELBO_{SMC}}$ with respect to $(\theta, \phi)$ for LGSSM and using the ALT algorithm for updating $(\theta, \phi)$ with $(A_\theta, K_\theta) = (\mathrm{SMC}, 1000)$ and $(A_\phi, K_\phi) = (\mathrm{IS}, 10)$. *Right* measures the quality of $\phi$ by showing $\sqrt{\sum_{t=1}^{T}(\mu_t^{\mathrm{kalman}} - \mu_t^{\mathrm{approx}})^2}$ where $\mu_t^{\mathrm{kalman}}$ is the marginal mean obtained from the Kalman smoothing algorithm under the model with EM-optimized parameters and $\mu_t^{\mathrm{approx}}$ is an marginal mean obtained from the set of 10 SMC particles with learned/bootstrap proposal.

## 5.3 MOVING AGENTS

To show that our results are applicable to complex, high dimensional data we compare AESMC and IWAE on stochastic, partially observable video sequences. Figure 7 in Appendix C.2 shows an example of such a sequence.

The dataset consists of $N = 5000$ sequences of images $(y_{1:T}^{(n)})_{n=1}^{N}$ of which 1000 are randomly held out as test set. Each sequence contains $T = 40$ images represented as a 2 dimensional array of size $32 \times 32$. In each sequence there is one agent, represented as circle, whose starting position is sampled randomly along the top and bottom of the image. The dataset is inspired by (Ondrúška & Posner, 2016), however with the crucial difference that the movement of the agent is *stochastic*. The agent performs a directed random walk through the image. At each timestep, it moves according to

$$y_{t+1} \sim \mathrm{Normal}(y_{t+1}; y_t + 0.15, 0.02^2)$$
$$x_{t+1} \sim \mathrm{Normal}(x_{t+1}; 0, 0.02^2) \tag{20}$$

where $(x_t, y_t)$ are the coordinates in frame $t$ in a unit square that is then projected onto $32 \times 32$ pixels. In addition to the stochasticity of the movement, half of the image is occluded, preventing the agent from being observed.

For the generative model and proposal distribution we use a Variational Recurrent Neural Network (VRNN) (Chung et al., 2015). It extends recurrent neural networks (RNNs) by introducing a stochastic

latent state $x_t$ at each timestep $t$. Together with the observation $y_t$, this state conditions the deterministic transition of the RNN. By introducing this unobserved stochastic state, the VRNN is able to better model complex long range variability in stochastic sequences. Architecture and hyperparameter details are given in Appendix C.1.

Figure 4 shows $\max(\mathrm{ELBO_{IS}}, \mathrm{ELBO_{SMC}})$ for models trained with IWAE and AESMC for different particle numbers. The lines correspond to the mean over three different random seeds and the shaded areas indicate the standard deviation. The same number of particles was used for training and testing, additional hyperparameter settings are given in the appendix. One can see that models trained using AESMC outperform IWAE and using more particles improves the ELBO for both. In Appendix C.2, we inspect different learned generative models by using them for prediction, confirming the results presented here. We also tested ALT on this task, but found that while it did occasionally improve performance, it was much less stable than IWAE and AESMC.

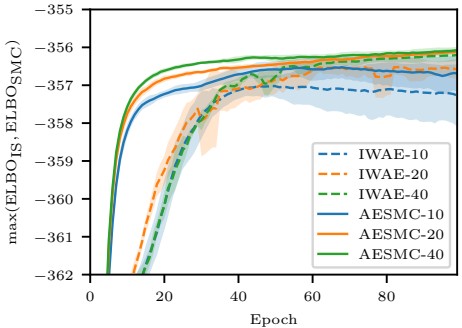

| Particles | Method | Moving Agents |
|---|---|---|
| 10 | IWAE | -357.3 |
| | AESMC | **-356.7** |
| 20 | IWAE | -356.6 |
| | AESMC | **-356.1** |
| 40 | IWAE | -356.2 |
| | AESMC | **-356.1** |

Figure 4: *(Left)* Rolling mean over 5 epochs of $\max(\mathrm{ELBO_{SMC}}, \mathrm{ELBO_{IS}})$ on the test set, lines indicate the average over 3 random seeds and shaded areas indicate standard deviation. The color indicates the number of particles, the line style the used algorithm. *(Right)* The table shows the final $\max(\mathrm{ELBO_{SMC}}, \mathrm{ELBO_{IS}})$ for each learned model.

# 6 CONCLUSIONS

We have developed AESMC—a method for performing model learning using a new ELBO objective which is based on the SMC marginal likelihood estimator. This ELBO objective is optimized using SGA and the reparameterization trick. Our approach utilizes the efficiency of SMC in models with intermediate observations and hence is suitable for highly structured models. We experimentally demonstrated that this objective leads to better generative model training than the IWAE objective for structured problems, due to the superior inference and tighter bound provided by using SMC instead of importance sampling.

Additionally, in Claim 1, we provide a simple way to express the bias of objectives induced by log of marginal likelihood estimators as a KL divergence on an extended space. In Propositions 1 and 2, we investigate the implications of these KLs being zero in the case of IWAE and AESMC. In the latter case, we find that we can achieve zero KL only if we are able to learn SMC intermediate target distributions corresponding to marginals of the target distribution. Using our assertion that tighter variational bounds are not necessarily better, we then introduce and test a new method, alternating ELBOs, that addresses some of these issues and observe that, in some cases, this improves both model and proposal learning.

ACKNOWLEDGMENTS

TAL is supported by EPSRC DTA and Google (project code DF6700) studentships. MI is supported by the UK EPSRC CDT in Autonomous Intelligent Machines and Systems. TR is supported by the European Research Council under the European Union's Seventh Framework Programme (FP7/2007-2013) ERC grant agreement no. 617071; majority of TR's work was undertaken while he was in the Department of Engineering Science, University of Oxford, and was supported by a BP industrial grant. TJ is supported by the UK EPSRC and MRC CDT in Statistical Science. FW is supported by The Alan Turing Institute under the EPSRC grant EP/N510129/1; DARPA PPAML through the U.S. AFRL under Cooperative Agreement FA8750-14-2-0006; Intel and DARPA D3M, under Cooperative Agreement FA8750-17-2-0093.

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

## A    GRADIENTS

The goal is to obtain an unbiased estimator for the gradient

$$\nabla_{\theta,\phi} \int Q_{\text{SMC}}(x_{1:T}^{1:K}, a_{1:T-1}^{1:K}) \log \hat{Z}_{\text{SMC}}(x_{1:T}^{1:K}, a_{1:T-1}^{1:K}) \, dx_{1:T}^{1:K} \, da_{1:T-1}^{1:K}. \tag{21}$$

### A.1    FULL REINFORCE

We express the required quantity as

$$\nabla_{\theta,\phi} \int Q_{\text{SMC}}(x_{1:T}^{1:K}, a_{1:T-1}^{1:K}) \log \hat{Z}_{\text{SMC}}(x_{1:T}^{1:K}, a_{1:T-1}^{1:K}) \, dx_{1:T}^{1:K} \, da_{1:T-1}^{1:K} \tag{22}$$

$$= \int \nabla_{\theta,\phi} Q_{\text{SMC}}(x_{1:T}^{1:K}, a_{1:T-1}^{1:K}) \log \hat{Z}_{\text{SMC}}(x_{1:T}^{1:K}, a_{1:T-1}^{1:K}) + \tag{23}$$

$$Q_{\text{SMC}}(x_{1:T}^{1:K}, a_{1:T-1}^{1:K}) \nabla_{\theta,\phi} \log \hat{Z}_{\text{SMC}}(x_{1:T}^{1:K}, a_{1:T-1}^{1:K}) \, dx_{1:T}^{1:K} \, da_{1:T-1}^{1:K} \tag{24}$$

$$= \int Q_{\text{SMC}}(x_{1:T}^{1:K}, a_{1:T-1}^{1:K}) \left[ \nabla_{\theta,\phi} \log Q_{\text{SMC}}(x_{1:T}^{1:K}, a_{1:T-1}^{1:K}) \log \hat{Z}_{\text{SMC}}(x_{1:T}^{1:K}, a_{1:T-1}^{1:K}) + \tag{25} \right.$$

$$\left. \nabla_{\theta,\phi} \log \hat{Z}_{\text{SMC}}(x_{1:T}^{1:K}, a_{1:T-1}^{1:K}) \right] \, dx_{1:T}^{1:K} \, da_{1:T-1}^{1:K}, \tag{26}$$

which we can estimate by sampling $(x_{1:T}^{1:K}, a_{1:T-1}^{1:K})$ directly from $Q_{\text{SMC}}$ and evaluating $\left[ \nabla_{\theta,\phi} \log Q_{\text{SMC}}(x_{1:T}^{1:K}, a_{1:T-1}^{1:K}) \log \hat{Z}_{\text{SMC}}(x_{1:T}^{1:K}, a_{1:T-1}^{1:K}) + \nabla_{\theta,\phi} \log \hat{Z}_{\text{SMC}}(x_{1:T}^{1:K}, a_{1:T-1}^{1:K}) \right]$.

### A.2    REINFORCE & REPARAMETERIZATION

We express the required quantity as

$$\nabla_{\theta,\phi} \int Q_{\text{SMC}}(x_{1:T}^{1:K}, a_{1:T-1}^{1:K}) \log \hat{Z}_{\text{SMC}}(x_{1:T}^{1:K}, a_{1:T-1}^{1:K}) \, dx_{1:T}^{1:K} \, da_{1:T-1}^{1:K} \tag{27}$$

$$= \nabla_{\theta,\phi} \int \left( \prod_{k=1}^{K} q_1(x_1^k) \right) \left( \prod_{t=2}^{T} \prod_{k=1}^{K} q_t(x_t^k | x_{t-1}^{a_{t-1}^k}) \cdot \text{Discrete}(a_{t-1}^k | w_{t-1}^{1:K}) \right)$$

$$\log \hat{Z}_{\text{SMC}}(x_{1:T}^{1:K}, a_{1:T-1}^{1:K}) \, dx_{1:T}^{1:K} \, da_{1:T-1}^{1:K} \tag{28}$$

$$= \nabla_{\theta,\phi} \int \left( \prod_{k=1}^{K} s_1(\epsilon_1^k) \right) \left( \prod_{t=2}^{T} \prod_{k=1}^{K} s_t(\epsilon_t^k) \cdot \text{Discrete}(a_{t-1}^k | w_{t-1}^{1:K}) \right)$$

$$\log \hat{Z}_{\text{SMC}}(r(\epsilon_{1:T}^{1:K}), a_{1:T-1}^{1:K}) \, d\epsilon_{1:T}^{1:K} \, da_{1:T-1}^{1:K} \tag{29}$$

$$= \int \left( \prod_{t=1}^{T} \prod_{k=1}^{K} s_t(\epsilon_t^k) \right) \left[ \nabla_{\theta,\phi} \prod_{t=2}^{T} \prod_{k=1}^{K} \text{Discrete}(a_{t-1}^k | w_{t-1}^{1:K}) \log \hat{Z}_{\text{SMC}}(r(\epsilon_{1:T}^{1:K}), a_{1:T-1}^{1:K}) + \right.$$

$$\left. \left( \prod_{t=2}^{T} \prod_{k=1}^{K} \text{Discrete}(a_{t-1}^k | w_{t-1}^{1:K}) \right) \nabla_{\theta,\phi} \log \hat{Z}_{\text{SMC}}(r(\epsilon_{1:T}^{1:K}), a_{1:T-1}^{1:K}) \right] \, d\epsilon_{1:T}^{1:K} \, da_{1:T-1}^{1:K}$$

$$\tag{30}$$

$$= \int \left( \prod_{t=1}^{T} \prod_{k=1}^{K} s_t(\epsilon_t^k) \right) \left( \prod_{t=2}^{T} \prod_{k=1}^{K} \text{Discrete}(a_{t-1}^k | w_{t-1}^{1:K}) \right) \cdot$$

$$\left[ \nabla_{\theta,\phi} \log \left( \prod_{t=2}^{T} \prod_{k=1}^{K} \text{Discrete}(a_{t-1}^k | w_{t-1}^{1:K}) \right) \log \hat{Z}_{\text{SMC}}(r(\epsilon_{1:T}^{1:K}), a_{1:T-1}^{1:K}) + \right.$$

$$\left. \nabla_{\theta,\phi} \log \hat{Z}_{\text{SMC}}(r(\epsilon_{1:T}^{1:K}), a_{1:T-1}^{1:K}) \right] \, d\epsilon_{1:T}^{1:K} \, da_{1:T-1}^{1:K}, \tag{31}$$

where $r(\epsilon_{1:T}^{1:K})$ denotes a sample with identical distribution as $x_{1:T}^{1:K}$ obtained by passing the auxiliary samples $\epsilon_{1:T}^{1:K}$ through the reparameterization function.  We can thus estimate

the gradient by sampling $\epsilon_{1:T}^{1:K}$ from the auxiliary distribution, reparameterizing and evaluating

$$\left[\nabla_{\theta,\phi} \log\left(\prod_{t=2}^{T}\prod_{k=1}^{K}\text{Discrete}(a_{t-1}^{k}|w_{t-1}^{1:K})\right)\log\hat{Z}_{\text{SMC}}(r(\epsilon_{1:T}^{1:K}), a_{1:T-1}^{1:K}) + \nabla_{\theta,\phi}\log\hat{Z}_{\text{SMC}}(r(\epsilon_{1:T}^{1:K}), a_{1:T-1}^{1:K})\right].$$

In Figure 5, we demonstrate that the estimator in (31) has much higher variance if we include the first term.

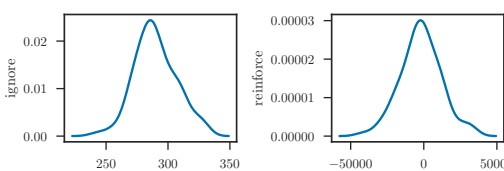

Figure 5: $T = 200$ model described in Section 5.1. Kernel density estimation (KDE) of $\nabla_{\theta_1} \text{ELBO}_{\text{SMC}}$ evaluated at $\theta_1 = 0.1$ with $K = 16$ using 100 samples.

## B  PROOFS FOR BIAS & IMPLICATIONS ON THE PROPOSALS

*Derivation of* (9).

$$\text{ELBO} = \int Q(x)\log\frac{Z_P P(x)}{Q(x)}\,\mathrm{d}x \tag{32}$$

$$= \int Q(x)\log Z_P\,\mathrm{d}x - \int Q(x)\log\frac{Q(x)}{P(x)}\,\mathrm{d}x \tag{33}$$

$$= \log Z_P - \text{KL}\left(Q||P\right). \tag{34}$$

□

*Proof of Claim 1.* Since $\hat{Z}_P(x) \geq 0$, $Q(x) \geq 0$ and $\int Q(x)\hat{Z}_P(x)\,\mathrm{d}x = Z_P$, we can let the unnormalized target density in Definition 1 be $\tilde{P}(x) = Q(x)\hat{Z}_P(x)$. Hence, the normalized target density is $P(x) = Q(x)\hat{Z}_P(x)/Z_P$. Substituting these quantities into (8) and (9) yields the two equalities in (10).

□

*Proof of Proposition 1.* ( $\implies$ ) Substituting for $Q_{\text{IS}}(x^{1:K}) = P_{\text{IS}}(x^{1:K})$, we obtain

$$\prod_{k=1}^{K} q(x^k|y) = \frac{1}{K}\sum_{k=1}^{K}\frac{\prod_{\ell=1}^{K}q(x^{\ell}|y)}{q(x^k|y)}p(x^k|y) \tag{35}$$

$$= \frac{1}{K}\sum_{k=1}^{K}\left[q(x^1|y)\cdots q(x^{k-1}|y)p(x^k|y)q(x^{k+1}|y)\cdots q(x^K|y)\right]. \tag{36}$$

Integrating both sides with respect to $(x^2, \ldots, x^K)$ over the whole support (i.e. marginalizing out everything except $x^1$), we obtain:

$$q(x^1|y) = \frac{1}{K}\left[p(x^1|y) + \sum_{k=2}^{K}q(x^1|y)\right]. \tag{37}$$

Rearranging gives us $q(x^1|y) = p(x^1|y)$ for all $x^1$.

( $\impliedby$ ) Substituting $p(x^k|y) = q(x^k|y)$, we obtain

$$P_{\text{IS}}(x^{1:K}) = \frac{1}{K}\sum_{k=1}^{K}\frac{Q_{\text{IS}}(x^{1:K})}{q(x^k|y)}p(x^k|y) \tag{38}$$

$$= \frac{1}{K}\sum_{k=1}^{K}Q_{\text{IS}}(x^{1:K}) \tag{39}$$

$$= Q_{\text{IS}}(x^{1:K}). \tag{40}$$

$\square$

*Proof of Proposition 2.* We consider the general sequence of target distributions $\pi_t(x_{1:t})$ ($p_\theta(x_{1:t}|y_{1:t})$ in the case of SSMs), their unnormalized versions $\gamma_t(x_{1:t})$ ($p_\theta(x_{1:t}, y_{1:t})$ in the case of SSMs), their normalizing constants $Z_t = \int \gamma_t(x_{1:t}) \, dx_{1:t}$ ($p_\theta(y_{1:t})$ in the case of SSMs), where $Z = Z_T = p(y_{1:T})$.

( $\implies$ ) It suffices to show that $\hat{Z}_{\text{SMC}}(x_{1:T}^{1:K}, a_{1:T-1}^{1:K}) = Z$ for all $(x_{1:T}^{1:K}, a_{1:T-1}^{1:K})$ implies 1 and 2 in Proposal 2 due to equation (11).

We first prove that $\hat{Z}_{\text{SMC}}(x_{1:T}^{1:K}, a_{1:T-1}^{1:K}) = Z$ for all $(x_{1:T}^{1:K}, a_{1:T-1}^{1:K})$ implies that the weights

$$w_1(x_1) := \frac{\gamma_1(x_1)}{q_1(x_1)} \tag{41}$$

$$w_t(x_{1:t}) := \frac{\gamma_t(x_{1:t})}{\gamma_{t-1}(x_{1:t-1})q_t(x_t|x_{1:t-1})} \qquad \text{for } t = 2, \ldots, T \tag{42}$$

are constant with respect to $x_{1:t}$.

Pick $t \in \{1, \ldots, T\}$ and distinct $k, \ell \in \{1, \ldots, K\}$. Also, pick $x_{1:t}$ and $x'_{1:t}$. Now, consider two sets of particle sets $(\bar{x}_{1:T}^{1:K}, \bar{a}_{1:T-1}^{1:K})$ and $(\tilde{x}_{1:T}^{1:K}, \tilde{a}_{1:T-1}^{1:K})$, illustrated in Figure 6, such that

$$\bar{x}_\tau^\kappa = \begin{cases} x'_\tau & \text{if } \kappa = \ell \text{ and } \tau < t \\ x'_\tau & \text{if } (\kappa, \tau) = (k, t) \\ x_\tau & \text{if } \kappa = k \text{ and } \tau < t \\ x_\tau^\kappa & \text{otherwise} \end{cases} \qquad \text{for } \tau = 1, \ldots, T, \; \kappa = 1, \ldots, K, \tag{43}$$

$$\bar{a}_\tau^\kappa = \begin{cases} \ell & \text{if } (\kappa, \tau) = (k, t-1) \text{ or } (k, t) \\ \kappa & \text{otherwise} \end{cases} \qquad \text{for } \tau = 1, \ldots, T-1, \; \kappa = 1, \ldots, K, \tag{44}$$

$$\tilde{x}_\tau^\kappa = \begin{cases} x'_\tau & \text{if } \kappa = \ell \text{ and } \tau < t \\ x_\tau & \text{if } (\kappa, \tau) = (k, t) \\ x_\tau & \text{if } \kappa = k \text{ and } \tau < t \\ x_\tau^\kappa & \text{otherwise} \end{cases} \qquad \text{for } \tau = 1, \ldots, T, \; \kappa = 1, \ldots, K, \tag{45}$$

$$\tilde{a}_\tau^\kappa = \begin{cases} \ell & \text{if } (\kappa, \tau) = (k, t) \\ \kappa & \text{otherwise} \end{cases} \qquad \text{for } \tau = 1, \ldots, T-1, \; \kappa = 1, \ldots, K. \tag{46}$$

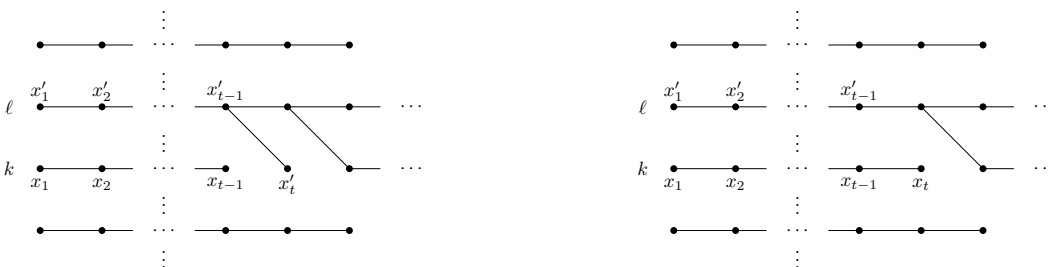

Figure 6: (Left) particle set $(\bar{x}_{1:T}^{1:K}, \bar{a}_{1:T-1}^{1:K})$ and (right) particle set $(\tilde{x}_{1:T}^{1:K}, \tilde{a}_{1:T-1}^{1:K})$. Lines indicate ancestor indices.

The weights $\bar{w}_\tau^\kappa$ and $\tilde{w}_\tau^\kappa$ for the respective particle sets are identical except when $(\tau, \kappa) = (t, k)$ where

$$\bar{w}_t^k = w_t(x'_{1:t}), \tag{47}$$

$$\tilde{w}_t^k = w_t(x_{1:t}). \tag{48}$$

Since $\hat{Z}(\bar{x}_{1:T}^{1:K}, \bar{a}_{1:T-1}^{1:K}) = \hat{Z}(\tilde{x}_{1:T}^{1:K}, \tilde{a}_{1:T-1}^{1:K})$, we have $w_t(x'_{1:t}) = w_t(x_{1:t})$. As this holds for any arbitrary $t$ and $x_{1:t}$, it follows that $w_t(x_{1:t})$ must be constant with respect to $x_{1:t}$ for all $t = 1, \ldots, T$.

Now, for $x_{1:t}$, consider the implied proposal by rearranging (41) and (42)

$$q_1(x_1) = \frac{\gamma_1(x_1)}{w_1} \tag{49}$$

$$q_t(x_t|x_{1:t-1}) = \frac{\gamma_t(x_{1:t})}{\gamma_{t-1}(x_{1:t-1})w_t} \qquad \text{for } t = 2, \ldots, T, \tag{50}$$

where $w_t := w_t(x_{1:t})$ is constant from our previous results. For this to be a normalized density with respect to $x_t$, we must have

$$w_1 = \int \gamma_1(x_1) \, dx_1 = Z_1, \tag{51}$$

and for $t = 2, \ldots, T$:

$$w_t = \int \frac{\gamma_t(x_{1:t})}{\gamma_{t-1}(x_{1:t-1})} \, dx_t \tag{52}$$

$$= \frac{\int \gamma_t(x_{1:t}) \, dx_t}{\gamma_{t-1}(x_{1:t-1})} \tag{53}$$

$$= \frac{Z_t}{Z_{t-1}} \cdot \frac{\int \pi_t(x_{1:t}) \, dx_t}{\pi_{t-1}(x_{1:t-1})}. \tag{54}$$

Since $\int \pi_{t+1}(x_{1:t+1}) \, dx_{t+1}$ and $\pi_t(x_{1:t})$ are both normalized densities, we must have $\pi_t(x_{1:t}) = \int \pi_{t+1}(x_{1:t+1}) \, dx_{t+1}$ for all $t = 1, \ldots, T-1$ for all $x_{1:t}$. For a given $t \in \{1, \ldots, T-1\}$ and $x_{1:t}$, applying this repeatedly yields

$$\pi_t(x_{1:t}) = \int \pi_{t+1}(x_{1:t+1}) \, dx_{t+1} = \int \int \pi_{t+2}(x_{1:t+2}) \, dx_{t+2} \, dx_{t+1} = \cdots = \int \pi_T(x_{1:T}) \, dx_{t+1:T} \tag{55}$$

such that each $\pi_t(x_{1:t})$ must be the corresponding marginal of the final target. We also have

$$w_1(x_1) = Z_1, \tag{56}$$

$$w_t(x_{1:t}) = \frac{Z_t}{Z_{t-1}}, \qquad\qquad t = 2, \ldots, T, \tag{57}$$

$$q_1(x_1) = \pi_1(x_1) = \pi_T(x_1), \tag{58}$$

$$q_t(x_t|x_{1:t-1}) = \frac{\pi_t(x_{1:t})}{\pi_{t-1}(x_{1:t-1})} = \frac{\pi_T(x_{1:t})}{\pi_T(x_{1:t-1})}, \qquad\qquad t = 2, \ldots, T. \tag{59}$$

( $\impliedby$ ) To complete the proof, we now simply substitute identities in 1 and 2 of Proposal 2 back to the expression of $\hat{Z}(x_{1:T}^{1:K}, a_{1:T-1}^{1:K})$ to obtain $\hat{Z}(x_{1:T}^{1:K}, a_{1:T-1}^{1:K}) = Z$. $\qquad\square$

## C    EXPERIMENTS

### C.1    VRNN

In the following we give the details of our VRNN architecture. The generative model is given by:

$$p(x_{1:T}, h_{0:T}, y_{1:T}) = p(h_0) \prod_t p(x_t|h_{t-1})p(y_t|h_{t-1}, x_t)p(h_t|h_{t-1}, x_t, y_t) \tag{60}$$

where

$$
\begin{aligned}
p(h_0) &= \text{Normal}(h_0; 0, I) \\
p(x_t|h_{t-1}) &= \text{Normal}(x_t; \mu_\theta^x(h_{t-1}), \sigma_\theta^x(h_{t-1})^2) \\
p(y_t|h_{t-1}, x_t) &= \text{Bernoulli}(y_t; \mu_\theta^y(\varphi_\theta^x(x_t), h_{t-1})) \\
p(h_t|h_{t-1}, x_t, y_t) &= \delta_{f(h_{t-1}, \varphi_\theta^x(x_t), \varphi_\theta^y(y_t))}(h_t)
\end{aligned}
\tag{61}
$$

and the proposal distribution is given by

$$p(x_t|y_t, h_{t-1}) = \text{Normal}(x_t; \mu_\phi^p(\varphi_\phi^y(y_t), h_{t-1}), \sigma_\phi^{p\,2}(\varphi_\phi^y(y_t), h_{t-1})) \tag{62}$$

The functions $\mu_\theta^x$ and $\sigma_\theta^x$ are computed by networks with two fully connected layers of size 128 whose first layer is shared. $\varphi_\theta^x$ is one fully connected layer of size 128.

For visual input, the encoding $\varphi_\theta^y$ is a convolutional network with conv-4x4-2-1-32, conv-4x4-2-1-64, conv-4x4-2-1-128 where conv-wxh-s-p-n denotes a convolutional network with $n$ filters of size $w \times h$, stride $s$, padding $p$. Between convolutions we use leaky ReLUs with slope 0.2 as nonlinearity and batch norms. The decoding $\mu_\theta^y$ uses transposed convolutions of the same dimensions but in reversed order, however with stride $s = 1$ and padding $p = 0$ for the first layer.

A Gated Recurrent Unit (GRU) is used as RNN and if not stated otherwise ReLUs are used in between fully connected layers.

For the proposal distribution, the functions $\mu_\phi^p$ and $\sigma_\phi^p$ are neural networks with three fully connected layers of size 128 that are sharing the first two layers. Sigmoid and softplus functions are used where values in $(0, 1)$ or $\mathbb{R}^+$ are required. We use a minibatch size of 25.

For the moving agents dataset we use ADAM with a learning rate of $10^{-3}$.

A specific feature of the VRNN architecture is that the proposal and the generative model share the component $\varphi_{\phi,\theta}^y$. Consequently, we set $\phi = \theta$ for the parameters belonging to this module and train it using gradients for both $\theta$ and $\phi$.

## C.2   MOVING AGENTS

In Figure 7 we investigate the quality of the generative model by comparing visual predictions. We do so for models learned by IWAE *(top)* and AESMC *(bottom)*. The models were learned using ten particles but for easier visualization we only predict using five particles.

The first row in each graphic shows the ground truth. The second row shows the averaged predictions of all five particles. The next five rows show the predictions made by each particle individually.

The observations (i.e. the top row) up to $t = 19$ are shown to the model. Up to this timestep the latent values $x_{0:19}$ are drawn from the proposal distribution $q(x_t|y_t, h_{t-1})$. From $t = 20$ onwards the latent values $x_{20:37}$ are drawn from the generative model $p(x_t|x_{t-1})$. Consequently, the model predicts the partially occluded, stochastic movement over 17 timesteps into the future.

We note that most particles predict a viable future trajectory. However, the model learned by IWAE is not as consistent in the quality of its predictions, often 'forgetting' the particle. This does not happen in every predicted sequence but the behavior shown here is very typical. Models learned by AESMC are much more consistent in the quality of their predictions.

## C.3   OPTIMIZING ONLY PROPOSAL PARAMETERS

We have run experiments where we optimize various ELBO objectives with respect to $\phi$ with $\theta$ fixed in order to see how various objectives have an effect on proposal learning. In particular, we train ELBO$_{\text{IS}}$ and ELBO$_{\text{SMC}}$ with number of particles $K \in \{10, 100, 1000\}$. Once the training is done, we use the trained proposal network to perform inference using both IS and SMC with number of particles $K_{\text{test}} \in \{10, 100, 1000\}$.

In Figure 8, we see experimental results for the LGSSM described in Section 5.1. We measure the quality of the inference network using a proxy $\sqrt{\sum_{t=1}^T (\mu_t^{\text{kalman}} - \mu_t^{\text{approx}})^2}$ where $\mu_t^{\text{kalman}}$ is the true marginal mean $\mathbb{E}_{p(x_{1:T}|y_{1:T})}[x_t]$ obtained from the Kalman smoothing algorithm and $\mu_t^{\text{approx}} = \left(\sum_{k=1}^K w_T^k x_t\right) / \left(\sum_{k=1}^K w_T^k\right)$ is an approximate marginal mean obtained from the proposal parameterized by $\phi$.

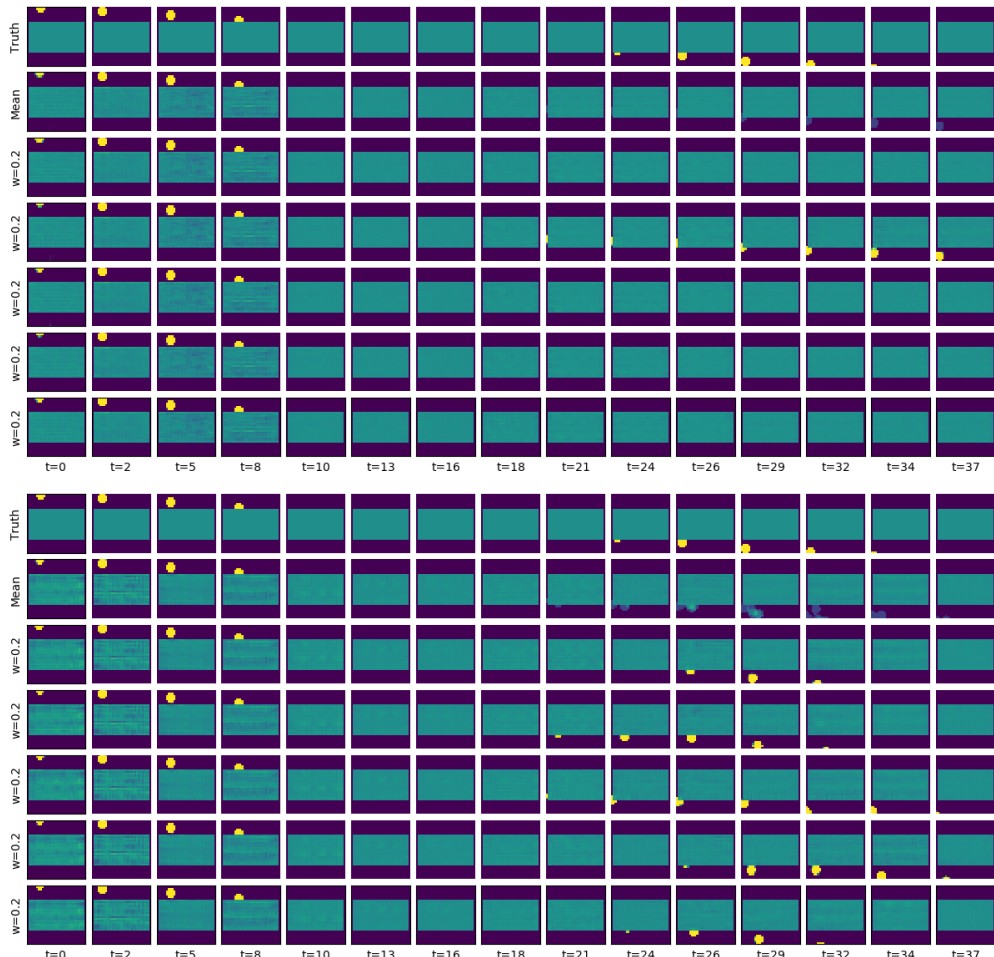

Figure 7: Visualisation of the learned model. Ground truth observations (top row in each sub figure) are only revealed to the algorithm up until t=19 inclusive. The second row shows the prediction averaged over all particles, all following rows show the prediction made by a single particle. *(Top)* IWAE. *(Bottom)* AESMC.

We see that if we train using $\text{ELBO}_{\text{SMC}}$ with $K_{\text{train}} = 1000$, the performance for inference using SMC (with whichever $K_{\text{test}} \in \{10, 100, 1000\}$) is worse than if we train with $\text{ELBO}_{\text{IS}}$ with any number of particles $K_{\text{train}} \in \{10, 100, 1000\}$. Examining the other axes of variation:

- Increasing $K_{\text{test}}$ (moving up in Figure 8 (Right)) improves inference.
- Increasing $K_{\text{train}}$ (moving to the right in Figure 8 (Right)) worsens inference.
- Among different possible combinations of (training algorithm, testing algorithm), (IS, SMC) $\succ$ (SMC, SMC) $\succ$ (IS, IS) $\succ$ (SMC, IS), where we use "$a \succ b$" to denote that the combination $a$ results in better inference than combination $b$.

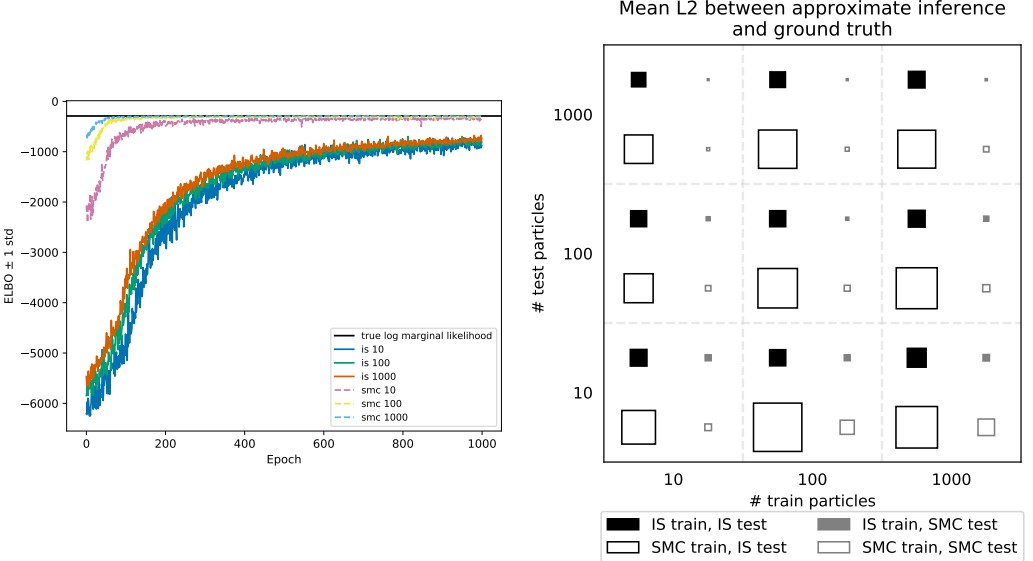

Figure 8: *(Left)* Optimizing ELBO with respect to $\phi$ for LGSSM. *(Right)* The lengths of the squares are proportional (with a constant factor) to $\sqrt{\sum_{t=1}^{T}(\mu_t^{\text{kalman}} - \mu_t^{\text{approx}})^2}$ which is a proxy for inference quality of $\phi$ described in the main text. The larger the square, the worse the inference.

