# OpenReview forum: "Auto-Encoding Sequential Monte Carlo"
_ICLR.cc/2018/Conference — Accept (Poster)_

### Official Review · AnonReviewer2 · 2017-11-21
**Auto-encoding Sequential Monte Carlo**

**Rating:** 3
**Confidence:** 2

**Review:**

Overall:
I had a really hard time reading this paper because I found the writing to be quite confusing. For this reason I cannot recommend publication as I am not sure how to evaluate the paper’s contribution.

Summary
The authors study state space models in the unsupervised learning case. We have a set of observed variables Y, we posit a latent set of variables X, the mapping from the latent to the observed variables has a parametric form and we have a prior over the parameters. We want to infer a posterior density given some data.

The authors propose an algorithm which uses sequential Monte Carlo + autoencoders. They use a REINFORCE-like algorithm to differentiate through the Monte Carlo. The contribution of this paper is to add to this a method which uses 2 different ELBOs for updating different sets of parameters.

The authors show the AESMC works better than importance weighted autoencoders and the double ELBO method works even better in some experiments.

The proposed algorithm seems novel, but I do not understand a few points which make it hard to judge the contribution. Note that here I am assuming full technical correctness of the paper (and still cannot recommend acceptance).

Is the proposed contribution of this paper just to add the double ELBO or does it also include the AESMC (that is, should this paper subsume the anonymized pre-print mentioned in the intro)? This was very unclear to me.

The introduction/experiments section of the paper is not well motivated. What is the problem the authors are trying to solve with AESMC (over existing methods)? Is it scalability? Is it purely to improve likelihood of the fitted model (see my questions on the experiments in the next section)?

The experiments feel lacking. There is only one experiment comparing the gains from AESMC, ALT to a simpler (?) method of IWAE. We see that they do better but the magnitude of the improvement is not obvious (should I be looking at the ELBO scores as the sole judge? Does AESMC give a better generative model?). The authors discuss the advantages of SMC and say that is scales better than other methods, it would be good to show this as an experimental result if indeed the quality of the learned representations is comparable.

---

> ### Comment · AnonReviewer1 · 2017-12-05
> **Regarding contribution**
>
> I have reviewed the paper as including the AESMC, so I would be interested in the answer to whether this is intended as well.

---

> ### Author Response · Authors · 2018-01-05
> **Response to AnonReviewer2**
>
> We thank the reviewer for taking the time to read through our paper and for their helpful feedback.
>
> We would like to reiterate the main contributions of the paper:
> - Re-introduction of the AESMC algorithm which was first introduced by Anon (2017) alongside the similar approaches of Maddison et al. (2017), and Naesseth et al. (2017).  We reiterate that our previous work Anon (2017) is only a preprint and so this work still constitutes the first introduction of AESMC to the literature.
>
> - Additional theoretical insights about the ELBOs used for AESMC and the IWAE, in particular demonstrating that increasing the number of particles K can be detrimental to proposal learning.
>
> - Introducing the alternating EBLO algorithm to ameliorate the problems about proposal learning that our theoretical insights highlight can occur for the original AESMC and IWAE algorithms.
>
> Regarding the comments about the experiments, we ran three experiments to illustrate our points:
> - The experiment described in section 5.1 provides evidence that the AESMC algorithm works on a time-series model where we know how to evaluate and maximize the log marginal likelihood exactly. Figure 2 demonstrates that AESMC works better than IWAE.
>
> - In section 5.2 we empirically investigate our claims about the adverse effect of increasing number of particles K on learning q(x|y) (Figure 3, left). We then run the ALT algorithm to ameliorate this effect on a time series data for which the experiments are in Figure 3 (middle, right).
>
> - Finally, we run both IWAE, AESMC and ALT on a neural network model where it is impossible to evaluate the log marginal likelihood exactly and we must resort to max(ELBO_IS, ELBO_SMC) as a proxy. This is a common practice in evaluating deep generative models.

---

### Official Review · AnonReviewer1 · 2017-11-27
**Interesting extension of IWAEs using SMC**

**Rating:** 7
**Confidence:** 4

**Review:**

[After author feedback]
I think the approach is interesting and warrants publication. However, I think some of the counter-intuitive claims on the proposal learning are overly strong, and not supported well enough by the experiments. In the paper the authors also need to describe the differences between their work and the concurrent work of Maddison et al. and Naesseth et al.

[Original review]
The authors propose auto-encoding sequential Monte Carlo (SMC), extending the VAE framework to a new Monte Carlo objective based on SMC. The authors show that this can be interpreted as standard variational inference on an extended space, and that the true posterior can only be obtained if we can target the true posterior marginals at each step of the SMC procedure. The authors argue that using different number of particles for learning the proposal parameters versus the model parameters can be beneficial.

The approach is interesting and the paper is well-written, however, I have some comments and questions:

- It seems clear that the AESMC bound does not in general optimize for q(x|y) to be close to p(x|y), except in the IWAE special case. This seems to mean that we should not expect for q -> p when K increases?
- Figure 1 seems inconclusive and it is a bit difficult to ascertain the claim that is made. If I'm not mistaken K=1 is regular ELBO and not IWAE/AESMC? Have you estimated the probability for positive vs. negative gradient values for  K=10? To me it looks like the probability of it being larger than zero is something like 2/3. K>10 is difficult to see from this plot alone.
- Is there a typo in the bound given by eq. (17)? Seems like there are two identical terms. Also I'm not sure about the first equality in this equatiion, is I^2 = 0 or is there a typo?
- The discussion in section 4.1 and results in the experimental section 5.2 seem a bit counter-intuitive, especially learning the proposals for SMC using IS. Have you tried this for high-dimensional models as well? Because IS suffers from collapse even in the time dimension I would expect the optimal proposal parameters learnt from a IWAE-type objective will collapse to something close to the the standard ELBO. For example have you tried learning proposals for the LG-SSM in Section 5.1 using the IS objective as proposed in 4.1? Might this be a typo in 4.1? You still propose to learn the proposal parameters using SMC but with lower number of particles? I suspect this lower number of particles might be model-dependent.

Minor comments:
- Section 1, first paragraph, last sentence, "that" -> "than"?
- Section 3.2, "... using which..." formulation in two places in the firsth and second paragraph was a bit confusing
- Page 7, second line, just "IS"?
- Perhaps you can clarify the last sentence in the second paragraph of Section 5.1 about computational graph not influencing gradient updates?
- Section 5.2, stochastic variational inference Hoffman et al. (2013) uses natural gradients and exact variational solution for local latents so I don't think K=1 reduces to this?

---

> ### Author Response · Authors · 2018-01-05
> **Response for AnonReviewer1**
>
> We thank the reviewer for taking the time to read through our paper and their insightful questions.
>
> Minor comments have been incorporated in the revision of our submission.
>
> We address the specific questions in turn:
>
> %%% It seems clear that the AESMC bound does not, in general, optimize for q(x|y) to be close to p(x|y)... %%%
>
> > This is true, but is effectively equivalent to the fact that the perfect importance sampler is better than the perfect SMC sampler, even though the latter is generally much more powerful when not perfectly optimized, as is almost always the case and hence why SMC is typically a superior inference algorithm. As we show in the set of equations (12-13) for IWAE and (14-15) for AESMC, these objectives can be decomposed to a log marginal likelihood term and a KL term on an extended space. In propositions 1 and 2, we prove that while we should expect q(x|y) = p(x|y) when we optimize the IWAE objective perfectly, this is not the case for AESMC.
>
>
> %%% Figure 1 seems inconclusive and it is a bit difficult to ascertain the claim that is made... %%%
>
> > The second paragraph of section 4 (and the associated Figure 1) is supposed to give an intuition for why using more K might result in a worse q(x|y). The case K=1 is regular ELBO (which is a special case of the IWAE) and the other cases are IWAE with the corresponding number of particles. We formalize this intuition by introducing the notion of a signal-to-noise ratio.
>
> The probabilities of the \grad_\phi ELBO estimator are estimate using 10000 Monte Carlo samples to be:
> K=1: 0.8704
> K=10: 0.6072
> K=100: 0.5226
> K=1000: 0.5036
> We believe that this is quite conclusive for this simple example, but the aim here is not the assumption that this simple example will generalize, just to show that increase K can be harmful.  However, our theoretical result on the signal-to-noise ratio does provide a generalization to general problems and shows that this effect must always manifest for sufficiently large K for any problem.
>
>
> %%% Is there a typo in the bound given by eq. (17)... %%%
>
> > Neither are typos.  The repeated term is because one gets an identical term in the bias and variance components of the bound, one of which goes to the numerator and one the denominator when we calculate the SNR.  It is indeed the case that I^2=0.
>
>
> %%% The discussion in section 4.1 and results in the experimental section 5.2 seem a bit counter-intuitive ... %%%
>
> > We agree that discussion and results with regards to the ALT algorithm are at first counter-intuitive, but we believe this is because of the counter-intuitive nature of our observation that increasing K actually harms the training of the proposal q(x|y).  Given this novel realization, using fewer particles to train the proposals becomes a natural thing to do, while our empirical results verify that it can lead to improvements in performance.
>
> We have indeed tried running ALT where we update \theta (generative parameters) using SMC with 1000 particles and \phi (proposal parameters) using IS with 10 particles. The results of this are described in the third paragraph of section 5.2. and the results are in Figure 3 (middle, right). We have also run the experiment on a neural network based model described in section 5.3 and Figure 4.

---

> > ### Comment · AnonReviewer1 · 2018-01-09
> > **Response to author feedback**
> >
> > Thank you for the clarifying comments, I will adjust my review accordingly.
> >
> > %%% It seems clear that the AESMC bound does not, in general, optimize for q(x|y) to be close to p(x|y)... %%%
> > You argue that increased K is detrimental to optimize q(z|x) to be close to p(z|x). But if the method is not designed to optimize the proposal to be close to the posterior, which you seem to agree with above, why should this be an issue?
> >
> > You show that for finite time the alternate seems to perform better, but do you have any results on whether this extends to when the high-sample version has actually converged? I'm referring to the LGSSM example, where the SMC-1000 hasn't yet converged. Paraphrased will the alternate still be better in the limit of infinite computation?
> >
> > This relates to another whether if it is possible that further gains on the alternate version can be achieved by an increasing schedule of particles for the proposal learning.
> >
> > %%% The discussion in section 4.1 and results in the experimental section 5.2 seem a bit counter-intuitive ... %%%
> > I think it would be very illustrative if there were experiments where the focus is actually on learning only \phi, instead of both \phi and \theta. Now the claim seems to be that learning proposals (\phi) with the IWAE  objective and a low number of particles is actually a better way of learning proposals for AESMC with a higher number of particles, rather than directly optimizing the AESMC objective. Could this also be related to the bias in the AESMC gradients?
> >
> > In Figure 4 you make mention to max(IS, SMC), in the experiments which one has been picked in each case? Does the ALT algorithm tend to pick one over the other?

---

> > > ### Author Response · Authors · 2018-01-16
> > > **Response to Response to author feedback**
> > >
> > > Thank you for the further comments.
> > >
> > > %%% You argue that increased K ... %%%
> > >
> > > >>> The argument about increasing K being detrimental to optimizing q(z|x) is actually distinct to the bound potentially not being tight.  Increasing K can be detrimental because of undermining our ability to reliably estimate the gradients, but is not detrimental in the true gradient direction itself.  The reason the bound does not become tight is that the “best” SMC proposal is generally not for q(x|y)=p(x|y). Thus is it still important to optimize q(x|y), it is just that this optimum is distinct to p(x|y).  Increasing K can still be detrimental to this process because of reducing the signal to noise ratio of the gradient estimates, thus harming our ability to learn the optimal SMC proposal.
> > >
> > > %%% You show that for finite time ... This relates to another ... %%%
> > >
> > > >>> We have run the experiment in the LGSSM example for more iterations (Figure 3 updated in latest revision) to characterize this further.  Our result now show that SMC-10 appears to be converging to a worse proposal that ALT (in terms of the marginal posterior), while SMC-1000 is still yet to converge.
> > > From the perspective of the theoretical limit of infinite computation in a generic stochastic gradient ascent scheme than our results on the SNR (which are the motivation for ALT) are inconsequential because they only relate to variance of the estimator. However, the assumptions required for this convergence are typically not satisfied for neural network training anyway, so the impact of the SNR may still be felt in the limit for real neural net training.  Moreover, this theoretical converge limit is typically very distinct to the point at which the training saturates - training can appear to have converged far more quickly than it truly has.
> > >
> > > Aside to our SNR arguments, there are also differences in the proposal achieved by ALT in the limit because of the effects previously discussed about the optimal importance sampling and SMC proposals being distinct (note the optimal SMC proposal also varies with K).  It can be hard to assess which of these is “better” because there is no objective metric for what a good q is.  For example, the KL tends to over-prefer low variance q’s for use as proposal distributions.  To try and investigate the relative metrics of different proposals, we have done an empirical investigation into training \phi for a given \theta as discussed next.
> > >
> > > %%% I think it would be very illustrative ... %%%
> > >
> > > >>> We have run additional experiments (included in a revision in Appendix E.1.) which confirm that if we want to optimize just \phi, then running IWAE with low number of particles is the best. In the experiment in Appendix E.1., we sweep through all possibilities of train_algorithm x train_number_of_particles x test_algorithm x test_number_of_particles where
> > >     - train_algorithm = test_algorithm = {IS, SMC}
> > >     - train_number_of_particles = test_number_of_particles = {10, 100, 1000}
> > >
> > > In the experiment, we confirm that using the IWAE objective with low number of particles results in a better \phi (in terms of inference) than optimizing AESMC with higher number of particles. Investigating other axes of variation:
> > >     - Inference performance worsens when we increase K_train.
> > >     - Inference performance improves when we increase K_test.
> > >     - The worst inference performance happens when we train with SMC with a lot of particles and test with IS with few particles.
> > >     - The best inference performance happens when we train with IS with few particles and test with SMC with a lot of particles.
> > >
> > > %%% Could this also be related to the bias in the AESMC gradients? %%%
> > >
> > > >>> As explained in our earlier comment, we think this is mostly tangential to the fact that the optimal q(x|y) for SMC is not the marginal posterior.
> > >
> > > %%% In Figure 4 you make mention to max(IS, SMC), in the experiments which one has been picked in each case? Does the ALT algorithm tend to pick one over the other?%%%
> > >
> > > >>> We currently don’t have data for this but will include it in a later revision.

---

### Official Review · AnonReviewer3 · 2017-12-05
**Some very interesting ideas and empirical results. But there are a couple of issues.**

**Rating:** 7
**Confidence:** 3

**Review:**

Update:
On further consideration (and reading the other reviews), I'm bumping my rating up to a 7. I think there are still some issues, but this work is both valuable and interesting, and it deserves to be published (alongside the Naesseth et al. and Maddison et al. work).

-----------

This paper proposes a version of IWAE-style training that uses SMC instead of classical importance sampling. Going beyond the several papers that proposed this simultaneously, the authors observe a key issue: the variance of the gradient of these IWAE-style bounds (w.r.t. the inference parameters) grows with their accuracy. They therefore propose using a more-biased but lower-variance bound to train the inference parameters, and the more-accurate bound to train the generative model.

Overall, I found this paper quite interesting. There are a few things I think could be cleared up, but this seems like good work (although I'm not totally up to date on the very recent literature in this area).

Some comments:

* Section 4: I found this argument extremely interesting. However, it’s worth noting that your argument implies that you could get an O(1) SNR by averaging K noisy estimates of I_K. Rainforth et al. suggest this approach, as well as the approach of averaging K^2 noisy estimates, which the theory suggests may be more appropriate if the functions involved are sufficiently smooth, which even for ReLU networks that are non-differentiable at a finite number of points I think they should be.

This paper would be stronger if it compared with Rainforth et al.’s proposed approaches. This would demonstrate the real tradeoffs between bias, variance, and computation. Of course, that involves O(K^2) or O(K^3) computation, which is a weakness. But one could use a small value of K (say, K=5).

That said, I could also imagine a scenario where there is no benefit to generating multiple noisy samples for a single example versus a single noisy sample for multiple examples. Basically, these all seem like interesting and important empirical questions that would be nice to explore in a bit more detail.

* Section 3.3: Claim 1 is an interesting observation. But Propositions 1 and 2 seem to just say that the only way to get a perfectly tight SMC ELBO is to perfectly sample from the joint posterior. I think there’s an easier way to make this argument:

Given an unbiased estimator \hat{Z} of Z, by Jensen’s inequality E[log \hat{Z}] ≤ log Z, with equality iff the variance of \hat{Z} = 0. The only way to get an SMC estimator’s variance to 0 is to drive the variance of the weights to 0. That only happens if you perfectly sample each particle from the true posterior, conditioned on all future information.

All of which is true as far as it goes, but I think it’s a bit of a distraction. The question is not “what’s it take to get to 0 variance” but “how quickly can we approach 0 variance”. In principle IS and SMC can achieve arbitrarily high accuracy by making K astronomically large. (Although [particle] MCMC is probably a better choice if one wants extremely low bias.)

* Section 3.2: The choice of how to get low-variance gradients through the ancestor-sampling choice seems seems like an important technical challenge in getting this approach to work, but there’s only a very cursory discussion in the main text. I would recommend at least summarizing the main findings of Appendix A in the main text.

* A relevant missing citation: Turner and Sahani’s “Two problems with variational expectation maximisation for time-series models” (http://www.gatsby.ucl.ac.uk/~maneesh/papers/turner-sahani-2010-ildn.pdf). They discuss in detail some examples where tighter variational bounds in state-space models lead to worse parameter estimates (though in a quite different context and with a quite different analysis).

* Figure 1: What is the x-axis here? Presumably phi is not actually 1-dimensional?

Typos etc.:

* “learn a particular series intermediate” missing “of”.

* “To do so, we generate on sequence y1:T” s/on/a/, I think?

* Equation 3: Should there be a (1/K) in Z?

---

> ### Author Response · Authors · 2018-01-05
> **Response to AnonReviewer3**
>
> We thank the reviewer for taking the time to review our paper and for their helpful feedback.
>
> %%% Relationship with Rainforth et al %%%
>
> > We would like to clarify that Rainforth et al do not propose any algorithmic approach, they only express the IWAE and VAE objectives in a general form and then carry out a theoretical analysis on which we build.  One can of course always use more Monte Carlo samples (i.e. increase M in their notation) to increase the fidelity of estimates.  The interesting approach which you are suggesting of using K^2 estimates here could be achieved by running K SMC sweeps of K samples.  We agree that this could be an interesting further extension on top of our suggested approach, but we did not have the time to actively investigate it for this revision.
>
>
> %%% Propositions 1 and 2 seem to just say that the only way to get a perfectly tight SMC ELBO is to perfectly sample from the joint posterior...  %%%
>
> > Your intuition here is correct - our proof relates to demonstrating this more formally and highlighting the fact that this requires a particular factorization of the model. Proving the “if” part of propositions 1, 2 is trivial.  However, proving the “only if” part of proposition 2 requires somewhat more care to show that the variance of the estimator can indeed only be zero when the variance of the weights is zero and that the variance of the weights can indeed only be zero if the intermediate targets incorporate all future information, implying a particular factorization of the generative model.
>
>
> %%% .... The question is not “what’s it take to get to 0 variance” but “how quickly can we approach 0 variance”... %%%
>
> > Though we agree with your sentiment that the speed of with 0 variance is approached is of critical importance and note that we provide empirical investigation of this through the experiments, we would like to reiterate that the key point of the result is to show that for 1<K<inf one will never learn a perfect estimator unless a particular factorization can be achieved.  This is at odds to cases where K=1, for which infinite training iterations should always lead to q becoming the posterior if q is expressive enough to encode it and the problem is convex.  In other words, in the IS case we can achieve exact posterior samples at test time with a finite K and a perfectly trained q for any model that can be represented by the inference network, but it the SMC case this is only possible if we also learn the optimal factorization of the model (as per proposition 2).
>
>
> %%% Section 3.2 %%%
>
> We believe that in practice, the bias introduced by ignoring this term is only very small.  We have added a short summary of the results from Appendix A as suggested.
>
>
> %%% Relevant missing citation: Turner and Sahani %%%
>
> > Thank you, we have duly updated the paper to include this reference.
>
>
> %%% Figure 1: What is the x-axis here? Presumably phi is not actually 1-dimensional?
>
> > In general, we are considering each dimension of the gradient separately in our assessment and so this should be read as \nabla_{\phi_1}.  Note that each dimension of the gradient being equally likely to be positive or negative corresponds to the overall gradient taking a completely random direction.
>
>
> %%% Typos etc. %%%
>
> * “learn a particular series intermediate” missing “of”.
>
> > Thank you, now fixed.
>
> * “To do so, we generate on sequence y1:T” s/on/a/, I think?
>
> > Thank you, now fixed.
>
> * Equation 3: Should there be a (1/K) in Z?
>
> > Thank you, now fixed.

---

> ### Author Response · Authors · 2018-01-16
> **Thank you**
>
> Thank you for your further consideration and bumping up your score.

---

### Decision · Program_Chairs · 2018-01-29
**ICLR 2018 Conference Acceptance Decision**

**Decision:**

Accept (Poster)

**Comment:**

This work develops importance weighted autoencoder-like training but with sequential Monte Carlo.  The paper is interesting, well written and the methods are very timely (there are two highly related concurrent papers -  Naesseth et al. and Maddison et al.).  Initially, the reviewers shared concerns about the technical details of the paper, but the authors appear to addressed those and two reviews have been raised accordingly.   There is one outlier review (two 7s and one 3).  The 3 is the least thorough and has the lowest confidence (2) so that review is being weighted accordingly.

This appears to be a timely and interesting paper that is interesting to the community and warrants publication at ICLR.

Pros:
- Well written and clear
- An interesting approach
- Neat technical innovations
- Generative deep models are of great interest to the community (e.g. Variational Autoencoders)

Cons:
- Could include a better treatment of recent related literature
- Leaves a variety of open questions about specific details (i.e. from the reviews)